# Immunogenicity of MultiTEP-Platform-Based Recombinant Protein Vaccine, PV-1950R, Targeting Three B-Cell Antigenic Determinants of Pathological α-Synuclein

**DOI:** 10.3390/ijms23116080

**Published:** 2022-05-29

**Authors:** Karen Zagorski, Gor Chailyan, Armine Hovakimyan, Tatevik Antonyan, Sepideh Kiani Shabestari, Irina Petrushina, Hayk Davtyan, David H. Cribbs, Mathew Blurton-Jones, Eliezer Masliah, Michael G. Agadjanyan, Anahit Ghochikyan

**Affiliations:** 1Department of Molecular Immunology, Institute for Molecular Medicine, Huntington Beach, CA 92647, USA; kzagorski@immed.org (K.Z.); gorchailyan@gmail.com (G.C.); ahov@immed.org (A.H.); tantonyan@immed.org (T.A.); 2Institute for Memory Impairments and Neurological Disorders, University of California Irvine, Irvine, CA 92697, USA; skianish@uci.edu (S.K.S.); ipetrush@uci.edu (I.P.); hdavtyan@uci.edu (H.D.); cribbs@uci.edu (D.H.C.); mblurton@uci.edu (M.B.-J.); 3Sue and Bill Gross Stem Cell Research Center, University of California Irvine, Irvine, CA 92697, USA; 4Laboratory of Neurogenetics, National Institute of Aging, National Institute of Health, Bethesda, MD 20814, USA; eliezer.masliah@nih.gov

**Keywords:** Parkinson’s disease, Alzheimer’s disease, MultiTEP platform, DNA and protein MultiTEP-based vaccines, immunogenicity, anti-α-synuclein antibodies, α-synuclein pathology

## Abstract

Parkinson’s disease (PD) and dementia with Lewy bodies (DLB) are characterized by the aberrant accumulation of intracytoplasmic misfolded and aggregated α-synuclein (α-Syn), resulting in neurodegeneration associated with inflammation. The propagation of α-Syn aggregates from cell to cell is implicated in the spreading of pathological α-Syn in the brain and disease progression. We and others demonstrated that antibodies generated after active and passive vaccinations could inhibit the propagation of pathological α-Syn in the extracellular space and prevent/inhibit disease/s in the relevant animal models. We recently tested the immunogenicity and efficacy of four DNA vaccines on the basis of the universal MultiTEP platform technology in the DLB/PD mouse model. The antibodies generated by these vaccines efficiently reduced/inhibited the accumulation of pathological α-Syn in the different brain regions and improved the motor deficit of immunized female mice. The most immunogenic and preclinically effective vaccine, PV-1950D, targeting three B-cell epitopes of pathological α-Syn simultaneously, has been selected for future IND-enabling studies. However, to ensure therapeutically potent concentrations of α-Syn antibodies in the periphery of the vaccinated elderly, we developed a recombinant protein-based MultiTEP vaccine, PV-1950R/A, and tested its immunogenicity in young and aged D-line mice. Antibody responses induced by immunizations with the PV-1950R/A vaccine and its homologous DNA counterpart, PV-1950D, in a mouse model of PD/DLB have been compared.

## 1. Introduction

Not only extracellular pathological amyloid-β (Aβ) but also cell-to-cell transmission of intracellular misfolded tau and α-Syn molecules are implicated in the progression of Alzheimer’s disease (AD) as well as various tauopathies and α-synucleopathies [1]. This propagation of disease-related misfolded proteins provides critical insights into the mechanism of pathological progression of different proteinopathies, including Parkinson’s disease (PD) and dementia with Lewy bodies (DLB). The reduction of accumulation of misfolded proteins by passive administrations of animal models with antibodies suggests that propagation is likely the central mechanism of progression of AD/PD/DLB [2,3,4,5,6,7,8,9,10]. Interestedly, preclinical data with immunotherapy partially supported clinical results obtained after administrations of diseased people with fully human or humanized monoclonal antibodies (mAb) specific to Aβ, tau, or α-Syn [11,12,13,14,15,16,17]. More specifically, mAb reduce/inhibit the accumulation of aggregated Aβ, tau, or α-Syn in the brains of passively vaccinated people but failed to significantly slow disease progression, likely because the treatment was initiated too late. These results support our long-standing tenet that antibodies specific to misfolded proteins could work as a preventive, not therapeutic, treatment. However, due to the complexity of neurodegenerative diseases, the cost, and the need for frequent (monthly) intravenous injections of asymptomatic people with high concentrations of monoclonal antibodies, passive vaccination is not practical as a preventive measure. By contrast, immunogenic active vaccines have been used as preventive measures for over a hundred years. Thus, to develop an immunogenic vaccine, we first created a universal vaccine platform, MultiTEP. MultiTEP can overcome self-tolerance in vaccinated individuals by activating both naïve and memory Th cells and can minimize the variability in immune responses due to HLA diversity in humans [18,19]. By attaching B cell epitopes of Aβ, tau, or α-Syn to the MultiTEP platform, we have developed vaccines that induce high titers of antibodies in inbred mouse models of AD/PD and outbred non-human primates possessing MHC class II gene polymorphism similar to humans.

Using MultiTEP platform technology, we developed four DNA vaccines targeting B-cell epitopes of hα-Syn spanning aa85-99 (PV-1947D), aa109-126 (PV-1948D), and aa126-140 (PV-1949D) separately, as well as all of three B-cell epitopes simultaneously (PV-1950D). Immunizations of wild-type mice [20] and a mouse model of PD/DLB [21] with these DNA vaccines resulted in the production of antibodies and significant reduction of the total and protein-kinase-resistant hα-Syn, as well as neurodegeneration in a sex-dependent manner. On the basis of the immunogenicity and efficacy data, we selected for future studies PV-1950D, targeting all three B-cell epitopes of α-Syn concurrently.

Although DNA vaccines offer advantages (being easy to manufacture, more stable, less expensive, etc.), setbacks include the requirement of special devices for the delivery of plasmid through the plasma and nuclear membranes of host cells (e.g., electroporation, gene gun, or needle-free syringes) [22]. Moreover, even with these delivery systems, DNA vaccine immunogenicity in humans has been significantly less than expected from preclinical studies in small animals and non-human primates [23,24], and, therefore, is not currently applicable for mass vaccination. This report focuses on developing a homologous recombinant protein vaccine, PV-1950R, and comparing the immunogenicity of PV-1950R formulated in Advax^CpG^ adjuvant (PV-1950R/A) with its DNA counterpart PV-1950D in the same mouse model of DLB/PD.

## 2. Results

### 2.1. Immunogenicity of Recombinant Protein Vaccine, PV-1950R/A, in Young and Old D-Line Mice

We immunized young and aged D-line mice with PV-1950R/A, a universal MultiTEP platform-based adjuvanted protein vaccine, and analyzed the humoral immune responses. In young mice, PV-1950R/A induced high titers of antibodies specific to all three B-cell epitopes of α-Syn, spanning aa85–99, aa109–126, and aa126–140 (Table 1). However, the endpoint titers of antibodies specific to the aa85–99 epitope were lower than antibodies specific to aa109–126 (ns) and significantly lower than antibodies specific to the aa126–140 epitope (*p* < 0.01). Titers of antibodies specific to full-length α-Syn were significantly higher than antibody titers to each epitope separately (*p* < 0.0001). Next, we tested the immunogenicity of PV-1950R/A in 12–14-month-old D line mice with established DLB/PD-like pathology. Like young mice, immunized aged animals generated antibodies specific to all three B-cell antigenic determinants of human α-Syn, and the response level to full-length α-Syn was significantly higher. However, unlike young mice, the old DLB/PD mice generated equal levels of antibodies specific to peptides spanning aa85–99, aa109–126, and aa126–140 (Table 1). Although antibody titers, in general, were very high in all mice, titers to peptides aa109–126 and aa126–140 and full-length α-Syn were significantly higher in young mice than in old mice. In summary, the PV-1950R/A vaccine induced high titers of antibodies in both young and old mouse models of DLB/PD. Still, it was less immunogenic in aged mice with established α-Syn pathology, likely due to age-related immunosenescence.

### 2.2. Comparison of Humoral Immune Responses in D-Line Mice Vaccinated with PV-1950D and PV-1950R/A

We recently reported the immunogenicity of the PV-1950D DNA (Figure 1A) vaccine in D-line mice, a model of PD/DLB-mimicking synucleinopathies, at 2–4 months of age at the start of immunization.

These published data showed that immunizations induced high titers of antibodies binding to three B-cell antigenic determinants of pathological α-Syn simultaneously, reducing total and protein-kinase-resistant α-Syn, as well as neurodegeneration [21]. In this study, we developed PV-1950R, a recombinant protein counterpart (Figure 1B) to the DNA vaccine and compared the humoral immune responses of PV-1950D and PV-1950R formulated in Advax^CpG^ adjuvant (PV-1950R/A). The results demonstrated that 100% of D line mice immunized with either DNA- or protein-based vaccines induced robust humoral immune responses (Figure 1D). Interestingly, and unexpectedly, mice immunized with PV-1950D induced a higher level of antibodies than mice vaccinated with the adjuvanted protein vaccine PV-1950R/A (*p* ≤ 0.01). However, variability of humoral immune responses in individual animals was also higher in the case of PV-1950D, likely because of the more complex administration method (e.g., immunization followed by electroporation with an AgilePulse device).

It is known that Th1-type pro-inflammatory immune responses are essential for the protection against viral infections and cancer, whereas Th2-type anti-inflammatory responses have generally been shown to inhibit autoimmune diseases. In mice, the production of IgG1 antibody is primarily induced by Th2 cytokines, whereas the production of IgG2a/c antibody reflects the involvement of Th1 cytokines [25,26,27,28,29]. Thus, we measured the production of IgG1, IgG2a/c, and IgG2b anti-α-Syn antibodies generated by PV-1950R/A (Figure 1D) and compared it with that in mice immunized with PV-1950D (Figure 1C). The data demonstrated that immunization with adjuvanted PV-1950R/A induced similar levels of IgG1 and IgG2b antibodies and a significantly lower level of IgG2a/c antibodies (*p* ≤ 0.0001). IgG1/IgG2a/c ratio was equal to 4, suggesting that vaccination with PV-1950R/A induces polarized humoral immune responses towards the anti-inflammatory Th2 phenotype. On the contrary, mice immunized with electroporation-mediated delivery of DNA vaccine, PV-1950D, produced an equal amount of IgG1 and IgG2a/c antibodies specific to hα-Syn (IgG1/IgG2a/c ratio = 0.87), suggesting Th1/Th2 mixed immune response.

We also measured the relative avidity of IgG antibodies purified from the sera of mice immunized with PV-1950R/A and PV-1950D vaccines. Data presented in Figure 2 indicate that immunizations with both vaccines induced the production of antibodies with comparable relative avidity. The area under the ROC curve was 0.6 (95% confidence interval), and the *P*-value was 0.3708. The concentration of sodium thiocyanate required to dissociate 50% of antibodies (half-maximal effective dose, ED50) generated by PV-1950D was only slightly lower (0.18 M) than that for antibodies generated by PV-1950R/A (0.25 M).

Previously, we reported that IgG antibodies generated in wild-type mice immunized with PV-1950D recognized LB in the brain sections from several DLB cases [20]. In this study, we analyzed the binding of IgG antibodies purified from sera of mice vaccinated with the PV-1950R/A and PV-1950D to the amygdala sections of brains from DLB/AD combined pathology case (#41-04) characterized at UCI Brain Bank and Tissue Repository. The staining with Amylo-Glo, anti-total tau mAb, and antibodies specific to three B cell epitopes of hα-Syn, purified from sera of mice vaccinated with PV-1950R/A and PV-1950D, is shown in Figure 3A,B. The data demonstrated that antibodies purified from sera of mice immunized with either PV-1950D or PV-1950R/A recognized pathological hα-Syn in the amygdala regions of the brains from DLB/AD cases. Importantly, antibodies generated with both vaccines recognized LBs in the same brain regions as localized tau tangles and amyloid plaques (Figure 3). Of note, and as expected, these antibodies specific to three B-cell epitopes (aa85–99, aa109–126, and aa126–140) of hα-Syn did not stain control brains from non-DLB/AD cases (data not shown).

## 3. Discussion

Nucleic acid vaccination provides a unique alternative immunization method; DNA constructs are relatively safe, cost- and time-efficient, and do not require toxic conventional adjuvants [22,30,31]. While recently, one DNA and two RNA vaccines for COVID-19 have been approved for emergency use, most vaccines for human use are still based on recombinant protein technology. These preventive adjuvanted vaccines induce strong humoral and cellular immune responses against various pathogens in humans [32]. This is why, in this study, we decided to develop a counterpart of PV-1950D, a recombinant protein vaccine, PV-1950R, and compare their immunogenicity in a mouse model of DLB/PD.

Our recently published results [20,21,33] and data presented above argue that both vaccines, on the basis of the MultiTEP platform, can induce high titers of anti-α-Syn IgG antibodies in young and old D line mice of both sexes. Importantly, either DNA- or recombinant-protein-based vaccines induced high titers of IgG antibodies that bind to hα-Syn with similar avidity. Nevertheless, vaccination of mice with PV-1950D generated Th1/Th2 mixed immune responses, while PV-1950R/A polarized humoral immune responses toward anti-inflammatory Th2 phenotype that might be beneficial for future human trials. These results are consistent with our previously published data in AD mice immunized with MultiTEP-based vaccines targeting Aβ and tau pathology separately or simultaneously [34,35,36,37].

It is known that hα-Syn pathology in AD is associated with a more aggressive disease course and accelerated cognitive dysfunction [38]. Our data supported this observation and demonstrated that IgG antibodies purified from mice vaccinated with PV-1950R/A and PV-1950D recognized pathological hα-Syn in amygdala (Amy) sections of brains from DLB/AD cases. Using confocal microscopy, we also demonstrated the regional co-occurrence of pathological α-Syn with pathological tau and Aβ plaques in brain sections from the DLB/AD case, although we had not observed any colocalization previously reported by [39].

Preclinical data reported here and in [20,21,33] on vaccines for PD/DLB along with published results with Aβ and tau vaccines for AD [34,36,40,41,42,43,44] suggest that preventive vaccines based on universal MultiTEP platform technology could (i) overcome self-tolerance by inducing Th cell responses to MultiTEP, but not to self-Aβ, tau, or α-Syn B-cell epitopes; (ii) induce therapeutically potent concentrations of antibodies in the majority of vaccinated subjects; (iii) minimize variability of immune responses due to HLA diversity in humans [18,19]; and (iv) augment antibody production through activation of not only naïve Th cells, but also pre-existing memory Th cells [44], which will be especially beneficial for elderly patients with a significant decrease in the subpopulation of naïve Th cells in the elderly [45,46]. We further hypothesized that antibodies raised in humans immunized with MultiTEP-based preventive vaccines could reduce/inhibit aggregation of misfolded proteins involved in the pathogenesis of DLB/PD/AD and delay the disease onset. Delaying the pathology of DLB/PD/AD, even for several years, would have profound clinical, societal, and financial ramifications for the entire population affected by these diseases, including the patients and their caregivers. Thus, we intend that the adjuvanted protein vaccine, after completion of the IND-enabling efficacy, CMC, and safety/toxicity studies, could be transferred to clinical trials.

## 4. Materials and Methods

### 4.1. Mice

Female and male mice of C57BL6/DBA2 background (H-2^b/d^) over-expressing transgenic (Tg) hα-Syn under the PDGF-β promoter (D Line) were used in this study. Mice were 2–4 months (young) or 12–14 months (old) old at the start of immunization. All animals were housed in a temperature- and light-cycle-controlled facility, and their care was under the guidelines of NIH and an approved IACUC protocol at UC Irvine.

### 4.2. Vaccines

The generation of the PV-1950D DNA vaccine was described previously [20]. Briefly, minigene encoding signal sequence, three copies of each B-cell epitopes aa85–99, aa109–126, and aa126–140 {3 × (aa126–140) plus 3 × (aa109–126) plus 3 × (aa85–99)} has been ligated with a gene encoding MultiTEP composed of a string of 12 foreign Th epitopes [40] and cloned into the pVAX1 vector (Invitrogen, Carlsbad, CA, USA). Immunization-grade plasmids were purified from E. coli by Aldevron (Madison, WI, USA), and the correct sequences of the generated plasmids were confirmed by nucleotide sequence analysis.

For the generation of PV-1950R recombinant protein, minigene *PV-1950* was amplified by PCR using forward 5′-TCCAGGTTCCCATATGGAAATGCCTTCTGAGGAAGG and reverse: 5′-CTCGAGTGATTCAGCGATGCTCAGGG primers and subcloned into the *E. coli* expression vector pET24a (Novagen, Cambridge, MA, USA) in frame with 6xHis-Tag at the C-terminus. The correct sequence of the generated plasmid (*PV-1950R*) was confirmed by nucleotide sequence analysis. The recombinant protein was purified from *E. coli* BL21 (DE3) cells transformed with *PV-1950R* plasmid as described [43,47] and analyzed in 10% Bis-Tris gel electrophoresis (NuPAGE Novex Gel, Invitrogen, CA, USA). The specificity of bands was confirmed by Western blot (WB) using polyclonal antibodies [20]. Endotoxin levels were measured using E-TOXATE kits, as recommended by the manufacturer (Sigma, St Louis, MO, USA).

### 4.3. Immunization of Mice

The immunization of mice with PV-1950D was described previously. Briefly, 2.5–3.5-month-old male (M) and female (F) mice (*n* = 6M/7F) were administered intramuscularly (30 µL) with 20 µg of vaccine, and immediately electrical pulses were applied using the AgilePulse^TM^ device from BTX Harvard Apparatus [20]. Two groups of female and male young (2–4-month-old, *n* = 4M/11F) and old (12–14-month-old, *n* = 5M/5F) D line mice were immunized intramuscularly (50 µL) with PV-1950R (20 µg protein/mouse/injection), formulated with Advax^CpG^ (Vaxine Pty Ltd., Adelaide, Australia) adjuvant at 1 mg/mouse/injection (designated as PV-1950R/A vaccine). Mice in both groups were immunized four times at weeks 0, 2, 6, and 10. The sera collected before injection (pre-bleed) and on the 14th day after the last immunization were used to analyze the humoral immune responses and purification of antibodies.

### 4.4. Detection of Antibody Titers and Isotypes

The endpoint titers of immunogen-induced antibodies were detected via ELISA, as described previously [33]. Briefly, the wells of the ELISA plates were coated with 1 µg/mL of hα-Syn protein (rPeptide) or the appropriate peptide (hα-Syn_85–99_, hα-Syn_109–126,_ and hα-Syn_126–140_) synthesized at Genscript. Serial dilutions (from 1:1000 to 1:2,430,000) of immune sera were added to the wells. After incubation and washing, an appropriate HRP-conjugated IgG, goat anti-mouse IgG (1:2500 Jackson ImmunoResearch Laboratories, West Grove, PA, USA), was used as a secondary antibody. Plates were incubated and washed, and the reaction was developed by adding 3,3′,5,5′ tetramethylbenzidine (Pierce, IL, USA) substrate solution and stopped with 2N H_2_SO_4_. The optical density was read at 450 nm (FilterMax F5). Endpoint titers were calculated as the reciprocal of the highest sera dilution that gave a reading three times above the background levels of binding of non-immunized sera at the same dilution (cutoff). HRP-conjugated anti-IgG1, IgG2a, IgG2c, IgG2b, and IgM-specific antibodies (Bethyl Laboratories, Inc., Montgomery, TX, USA) were used to characterize the isotype profiles of anti-hα-Syn antibodies in individual sera collected after the fourth immunization at dilutions of 1:1000, 1:3000, 1:9000, and 1:27,000.

### 4.5. Measuring Avidity of Antibodies

*Purification of anti-hα-Syn antibodies:* Anti-hα-Syn antibodies from sera of mice immunized with PV-1950D and PV-1950R/A vaccines were purified by an affinity column (SulfoLink, ThermoFisher Sci., Waltham, MA, USA) using an immobilized α-Syn_85–99_-C, α-Syn_109–126_-C, and α-Syn_126–140_-C peptides (GenScript Company, Piscataway, NJ, USA). Sera were divided into three equal volumes. Each aliquot was purified by a column immobilized with one of three peptides, according to manufacturer recommendations. Purified antibodies were analyzed via 10% Bis-Tris gel (ThermoFisher Sci.), and the concentrations were determined using a BCA protein assay kit (ThermoFisher Sci.). Equal amounts of α-Syn_85–99_, α-Syn_109–126,_ and α-Syn_126–140_ antibodies were mixed for avidity measurement.

*Binding avidity:* Sodium thiocyanate (NaSCN) displacement ELISAs were performed by the method described by Richmond et al. [48]. Concentrations of purified antibodies from the sera of mice immunized with PV-1950D and PV-1950R/A were equalized to 80 µg/mL. Bound antibody was detected as described above, except that after incubation with primary antibodies, plates were washed three times with TTBS (TBS buffer containing 0.5% Tween 20); incubated with TBS buffer containing 0, 0.5, 1, 1.5, 2, 2.5, and 3.5 M NaCSN for 15 min; and then washed six more times with TTBS. The results were expressed as the percentage of the concentration of antibodies in the absence of NaSCN. The half-maximal effective dose (ED_50_) was calculated as the concentration of NaCSN required for the release of 50% of antibodies from the ELISA plate.

### 4.6. Detection of hα-Syn in Human Brain Tissues from AD/DLB Cases

The ability of antibodies to bind Lewy bodies (LB) in the human brain tissues and the colocalization of LB with amyloid plaques and tau tangles was analyzed by immunofluorescence, as we previously described [34]. Briefly, 40 µm brain sections of formalin-fixed frontal cortical (FCx) and amygdala (Amy) tissues from typical AD/DLB case was obtained from the UCI MiND Brain Bank and Tissue Repository with the following neuropathology: tangle stage 6, Aβ plaque stage C, LB detected in substantia nigra, amygdala, frontal/temporal/parietal/rostral cingulate, locus coeruleus, and innominate substance. After pretreatment with citrate buffer, sections were stained with Amylo-Glo^TM^ RTD Amyloid Plaque Stain Reagent (Biosensis, Thebarton, Australia) (1:100 dilution), then with rabbit anti-hTau mAb (Abcam, 1:250 dilution) and α-synuclein (2 μg/mL) primary antibodies purified from sera of mice immunized with PV-1950D and PV-1950R/A. Sections were then incubated in conjugated secondary antibodies (anti-mouse-488 and anti-rabbit-555 at 1:400 dilutions, Invitrogen) and mounted on microscope slides. Immunofluorescent sections were then visualized and captured using an Olympus FX 1200 confocal microscope, with identical laser and detection settings across a given immunolabel. LB, Tau tangles, and Aβ-plaques were visualized using Z-stack maximum-projection images taken through the entire depth of the section at 1 µm intervals.

### 4.7. Statistical Analysis

Statistical parameters (mean, standard deviation (SD), significant difference, etc.) were calculated using the Prism 9.3.1 software (GraphPad Software, Inc., San Diego, CA, USA). Statistically significant differences were examined using analysis of variance (ordinary one-way ANOVA) and Tukey’s multiple comparisons post-test or unpaired *t*-test (a *p*-value of less than 0.05 was considered significant).

## 5. Conclusions

In sum, in this study, we tested the immunogenicity of DNA and recombinant protein counterparts of MultiTEP-platform-based vaccines, targeting three B-cell epitopes of α-Syn. Both vaccines induced the production of high titers of antibodies in hα-Syn/Tg D line mice. The DNA-based vaccine induced a mixed Th2/Th1 phenotype, while the adjuvanted protein vaccine induced the Th2 phenotype of immune responses. Antibodies produced by both vaccines are bound to Lewy bodies in brain sections of patients with AD/DLB.

## Figures and Tables

**Figure 1 ijms-23-06080-f001:**
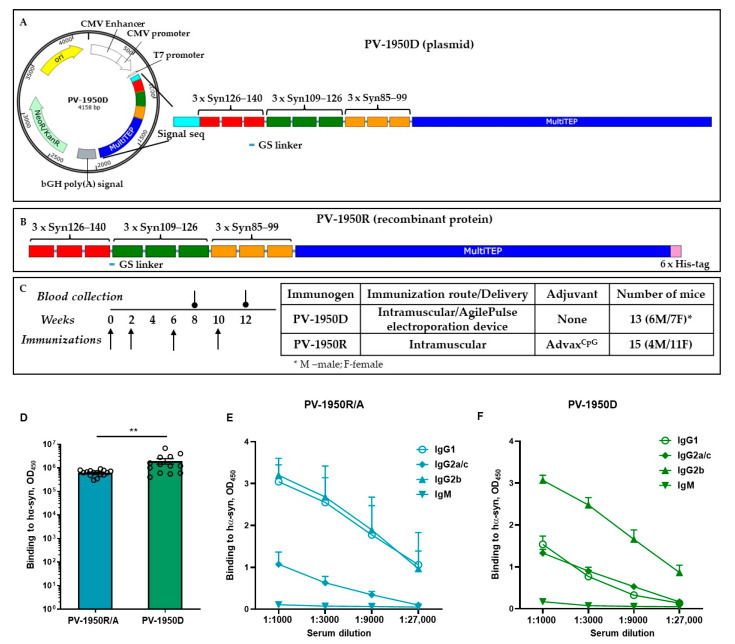
Schematic representation of (**A**) PV-1950D plasmid and (**B**) PV-1950R recombinant protein vaccines. (**C**) Schedule of immunization of hα-Syn Tg D line mice with DNA- and recombinant-protein-based vaccines. PV-1950D plasmid DNA was injected intramuscularly, followed by electrical pulses with an AgilePulse electroporation device. PV-1950R recombinant protein was formulated with Advax^CpG^ adjuvant (PV-1950R/A) and injected intramuscularly. (**D**) Titers of anti-Syn antibodies induced by PV-1950R/A (*n* = 13) and PV-1950D (*n* = 15) MultiTEP-based epitope vaccines in hα-Syn Tg D line mice (** *p* ≤ 0.01). (**E**) PV-1950R/A induced equal amounts of IgG1 and IgG2b and significantly lower amounts of IgG2a/c (*p* < 0.0001) and IgM (*p* < 0.0001) antibodies; the mean ratio IgG1/IgG2a/c was 4. (**F**) PV-1950D induced similar levels of IgG1 and IgG2a/c antibodies, while the level of IgG2b antibodies was significantly (*p* ≤ 0.0001) higher, and the level of IgM was significantly lower (*p* < 0.0001); the mean ratio IgG1/IgG2a/c was 0.87. Isotypes of antibodies were detected at indicated dilutions of sera collected from individual mice after the fourth immunization. Statistical analysis of isotypes was performed using data obtained from a 1:3000 dilution of sera using one-way ANOVA. Error bars indicate the mean values of antibody titers ± SEM.

**Figure 2 ijms-23-06080-f002:**
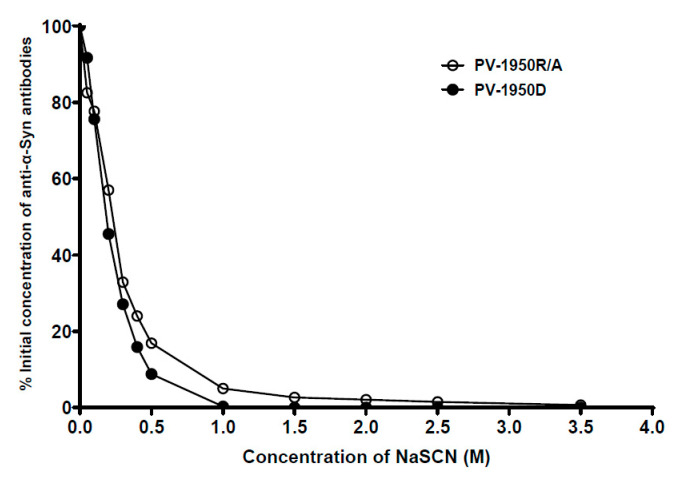
PV-1950R/A vaccine induced antibodies with slightly higher avidity for binding with hα-Syn protein than PV-1950D. Relative avidity for antibody–antigen binding was determined using sodium thiocyanate (NaSCN) displacement enzyme-linked immunosorbent assay (ELISA). The effective concentration of NaSCN required to release 50% of antiserum from the ELISA plate (half-maximal effective dose (ED50)) was 0.25 M for PV-1950R/A and 0.18 M for PV-1950D. ROC AUC score −0.6, *P* = 0.3708. The concentration of antibodies was 80 ng/mL.

**Figure 3 ijms-23-06080-f003:**
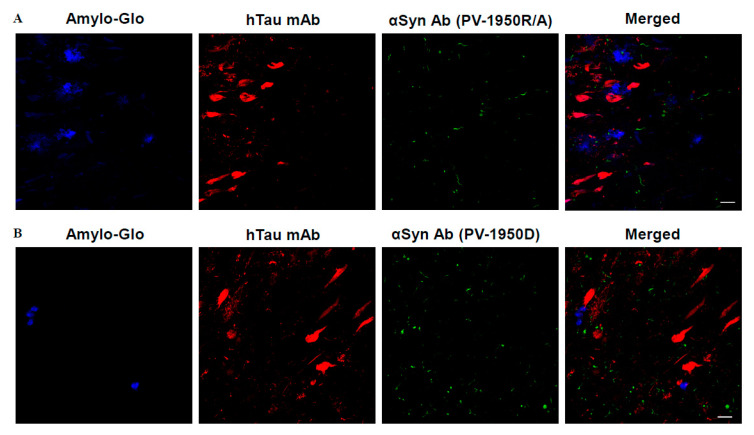
IgG antibodies induced by PV-1950R/A (**A**) and PV-1950D (**B**) vaccines in D line hα-Syn Tg mice recognized LBs in the brains from DLB/AD cases. Co-staining of the brain sections with anti-α-Syn antibodies induced either by PV-1950R/A or PV-1950D (green), Amylo-Glo (blue), and rabbit anti-total hTau antibodies (red), showing the presence of LB, amyloid plaques, and tau tangles in amygdala sections of DLB/AD brain, respectively. Magnification 40×, scale bar 20 μm.

**Table 1 ijms-23-06080-t001:** PV-1950R/A adjuvanted protein vaccine is immunogenic in young and aged D line mouse model of PD/DLB.

Age of D-Line Mice at the Start of Immunization ^#^	Endpoint Titers (±SEM) of Antibodies Binding to Three B cell Epitopes of α-Syn and Full-Length α-Syn			
	aa 85-99	aa 109-126	aa 126-140	α-Syn
2-4 mo old	1:16236 ±6 860	1:61000 ± 11685	1:179600 ± 43634	1:616667 ± 43124
12-14 mo old	1:11865 ± 5109	1:10453 ± 4196	1:12082 ± 7686	1:231000 ± 25438
*p* value	0.6312	0.0016	0.0035	<0.0001

^#^ Mice of both sexes, after vaccinations with PV-1950R/A, generated similar levels of anti-hα-Syn antibodies.

## Data Availability

All data generated or analyzed during this study are included in this article, and materials are available from the corresponding author on reasonable request.

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
