# Peer review of "Immunogenicity of MultiTEP-Platform-Based Recombinant Protein Vaccine, PV-1950R, Targeting Three B-Cell Antigenic Determinants of Pathological α-Synuclein"

_ijms, 2022, doi:10.3390/ijms23116080_

Round 1

Author Response

Q1. Title: It isn't clear from the title which vaccine PV-1950R is. Suggest changing it to "Immunogenicity of MultiTEP platform-based recombinant protein vaccine, PV-1950R, targeting three B cell antigenic determinants of pathological α-Synuclein."

 R1. We amended the title as recommended by the referee.

Q2. Intro: End of paragraph 1 needs improvement to conform to standard English.

 R2. We amended the end of paragraph 1 to conform to standard English as follows:

"Thus, to develop an immunogenic vaccine, we first created a universal vaccine platform, MultiTEP. MultiTEP can overcome self-tolerance in vaccinated individuals by activating both naïve and memory  Th cells and can minimize the variability in immune responses due to HLA diversity in humans [18,19]. By attaching B cell epitopes of Aβ, tau, or α-Syn to the MultiTEP platform, we have developed vaccines that induce high titers of antibodies in inbred mouse models of AD/PD and outbred non-human primates possessing MHC class II genes polymorphism similar to humans." Page 2.

Q3. Paragraph 1, 1950R/A needs to be defined. "Like young mice, aged animals induced antibodies" is incorrect English. "Induced" should be replaced by generated or developed throughout the manuscript when referring to mice.

R3. PV-1950R/A is defined (page 2), and "induced" is replaced by "generated" throughout the revised manuscript.

 Q4. Either 'aa 109-126' or 'aa109-126' should be used consistently throughout the manuscript.

R4. "aa109-126" is used consistently throughout the revised manuscript.

 Q5. Regarding the statement "antibodies specific to aa85-99 were significantly lower than antibodies specific to the other two B cell epitopes of α-Syn": In Table 1 the p values appear to be generated from comparisons between young and aged mice. Statistics (ANOVA) comparing the response to aa85-99 to the other epitopes needs to be performed if the authors wish to draw this Conclusion. The same is true of other comparisons where a significant difference is claimed.

R5. We performed statistical analysis (ANOVA) of titers of antibodies specific to epitopes aa 85-99, aa 109-126, and aa 126-140 and added p values into the text in the revised manuscript (pages 2-3). 

 Q6. If comparisons are indeed being made between young and aged mice for each epitope independently, then Students t test is more appropriate than the ANOVA analysis conducted. ANOVA would be appropriate if multiple epitopes are compared to each other or full-length.

 R6. Statistical analysis was performed using the Student t-test to compare titers of antibodies to each epitope and full-length α-Syn generated in young vs. old mice. Table 1 presents titers of antibodies and p values.

Q7. Paragraph 1 discusses studies with the 1950R/A vaccine but data in Table 1 are for AV-1959R/A vaccine. This inconsistency should be resolved and AV-1959R/A should be defined in this paragraph. Furthermore, the legend to Table 1 suggests male and female mice were immunized with PV-1950R/A instead?

R7. We apologize for this typo. This is corrected in the revised manuscript.

 Q8. What is the difference between PV-1950R and PV-1950R/A?

R8. PV-1950R is a recombinant protein, while PV-1950R/A is the PV-1950R formulated in AdvaxCpG adjuvant. PV-1950R/A is defined in the revised manuscript (page 2).

 Q9. Experimentally, isotype-specific antibody responses in Fig. 1b were measured only from a single dilution of serum. This resulted in one-third of the antibody titers being beyond the linear range of the assay. The serum should be re-run at several dilutions (like the assay to whole protein) so that an accurate, quantitative comparison can be made.

 R9. As recommended by the reviewer, we analyzed isotypes of generated antibodies using several dilutions of serum (1:1000, 1:3000, 1:9000, and 1:27000). New data are included in the revised manuscript.

 Q10. The legend in Fig. 1 contains many conclusions. It should be rewritten to merely describe what is being shown in the graphs.

R10. The legend for Fig.1 is amended and now describes what is shown in the graph without the conclusions.

 Q11. Fig. 2: Area under the curve statistical analysis should be performed to support the authors' Conclusion that the relative avidity was comparable.

R11. As recommended by the reviewer, we performed "Area under the curve" analyses for Figure 2 and included these data in the revised manuscript (page 4).

 Q12. Fig. 3 and its legend are very difficult to decipher. Do the authors mean to say that "Co-staining of these tissue sections with Amylo-Glo and rabbit anti-total hTau antibodies showed the presence of LB, amyloid plaques, and tau tangles in amygdala sections of DLB/AD brain" rather than "Co-staining of these antibodies with Amylo-Glo and rabbit anti-total hTau antibodies showed the presence of LB, amyloid plaques, and tau tangles in amygdala sections of DLB/AD brain"?

R12.  Fig. 3 legend is amended to make it more apparent in the revised manuscript.

 Q13. Discussion: In paragraph 2, the authors cannot conclude, based on the data presented, that both vaccines induce therapeutically potent antibodies. They merely demonstrated generation of antibodies of similar titer and avidity. Indeed, the authors demonstrated that the isotype ratios of the antibodies generated by the two vaccines differ, underscoring the need to directly assess therapeutic potential before drawing such a sweeping conclusion.

R13. We agree with the comment of the reviewer. We showed the therapeutic efficacy of antibodies generated by PV-1950D in the D line mouse model in our previous publication, cited in this manuscript. We are currently initiating the testing of the therapeutic efficacy of the PV-1950R/A vaccine, which will take an additional year, and the data will be presented as a separate manuscript. Therefore, we removed this statement from the revised manuscript (page 6).

 Q14. Conclusions: The hypothesis stated here belongs in the Discussion but is not appropriate as a conclusion to the present body of data.

R14. The hypothesis stated in the Conclusion was slightly changed and transferred to Discussion as recommended by the reviewer (page 6).

Reviewer 2 Report

Manuscript IJMS-1720909

Title: Immunogenicity of MultiTEP platform based recombinant vaccine, PV-1950R targeting three B cell antigenic determinants of pathological α-Synuclein

The main objective of this work is the evaluation of the vaccines AV-1950R/A and AV-1959R/A in a mouse model of Parkinson´s disease (PD) and Dementia with Lewy bodies (DLB).

The work, in general, is well written, of easy lecture, very interesting and without gross errors, but I would like to make the following considerations.

1.- Order the superscripts, putting # first. Finally, put the asterisk with the information of corresponding authors and senior authors

2.- Inconsistency between the acronym MAG (for Michael G. Agadjanyan) in the correspondence and in the Author Contributions (MGA)

3.- Correct typography throughout the text to match the size of the words.

4.- Please, include the full name of the IND in the paragraph “.., has been selected for the future IND enabling studies.”

5.- Please include amyloid beta before AB

6.- Figure 1. Please, can you include the number of antibody determinations (n)

In conclusion, this is a very interesting work can be improved by some little changes.

Author Response

Q1. Order the superscripts, putting # first. Finally, put the asterisk with the information of corresponding authors and senior authors

 R1. As recommended by the reviewer, we ordered the superscripts and put different symbols for the information of corresponding authors and senior authors.

 Q2. Inconsistency between the acronym MAG (for Michael G. Agadjanyan) in the correspondence and in the Author Contributions (MGA).

R2. We apologize for the typo. We corrected the typo and put MGA in the correspondence.

Q3. Correct typography throughout the text to match the size of the words.

R3. We corrected the size throughout the text.

 Q4. Please, include the full name of the IND in the paragraph ".., has been selected for the future IND enabling studies."

R4. We included the full name of IND (Investigational New Drug) in the revised manuscript.

 Q5. Please include amyloid beta before AB

R5. We included amyloid-β before first mentioning Aβ.

 Q6. Figure 1. Please, can you include the number of antibody determinations (n)

R6. The numbers of antibody determinations are included in the Figure 1 legend.